# A Hybrid Paradigm for Vision Autoencoders: Unifying CNNs and Transformers for Learning Efficiency and Scalability

## Abstract

Architectures for visual Variational Autoencoders (VAEs) have been dominated by Convolutional Neural Networks (CNNs), which inherently struggle to model long-range dependencies efficiently. While Vision Transformers (ViTs) offer a promising global receptive field, their direct application to VAEs has been hampered by a critical weakness in modeling fine-grained local details, leading to significant learning inefficiencies. This has left the field at an architectural impasse, limiting progress in high-fidelity representation learning. To break this impasse, we propose TransVAE, a hybrid paradigm that unifies a shallow CNN front-end for robust local feature extraction with a deep Transformer backbone for powerful global context modeling. TransVAE demonstrates superior **learning efficiency**, converging faster than CNN baselines while achieving state-of-the-art results. Critically, the comprehensive visual representation from our hybrid architecture unlocks three properties: First, **scalability** with performance consistently improving as parameters scale from 44M to 2.3B-a feat not effectively achieved by pure ViT VAEs. Second, enhanced **extrapolation** allows models trained on low resolutions to perform inference on arbitrary higher resolutions with superior global coherence. Third, a better **harmonization of pixel-level and semantic-level representation** facilitates both reconstruction and generation. TransVAE thus provides a new, effective blueprint for the next generation of visual VAEs. Codes and weights are available upon acceptance.

## 1 Introduction

Since the era of the latent diffusion model (Rombach et al., 2022), modern generative models adopt visual tokenizers to reduce the computational cost by compressing large-resolution visual inputs to a low-dimensional latent space. Among different tokenizers, the continuous-value variational autoencoder (VAE) (Kingma, 2013) stands out, as it enables high-fidelity encoding and decoding of visual information. For years, VAE architectures have been dominated by CNNs (Rombach et al., 2022; Esser et al., 2024; Labs). This paradigm has been successful, leveraging strong inductive biases such as locality and translation equivariance to efficiently learn hierarchical features from pixel-level data (Krizhevsky et al., 2012; He et al., 2016; Van Den Oord et al., 2017; Vahdat & Kautz, 2020). Despite their success, CNNs face a fundamental limitation: the local nature of the convolution operator makes it inefficient at modeling long-range, non-local dependencies. Capturing the global structure of a scene often requires deep networks, leading to computational and optimization challenges.

The rise of Vision Transformers (ViTs) presents a potential alternative, offering a native mechanism for global context modeling through their self-attention layers (Dosovitskiy et al., 2020). However, their promise has not fully translated to the VAE setting. The lack of strong local inductive biases makes ViTs inefficient at modeling fine-grained, high-frequency details from scratch. This critical weakness often results in blurry reconstructions and wrong pixel color, as shown in Figure 1.

Consequently, the field has reached an architectural impasse. Neither a purely local nor a purely global approach provides a complete solution for high-fidelity visual VAEs. A more profound issue arises when attempting to scale these monolithic architectures: simply scaling the parameter count in either pure CNN or pure ViT models fails to produce the expected improvements in quality. We attribute

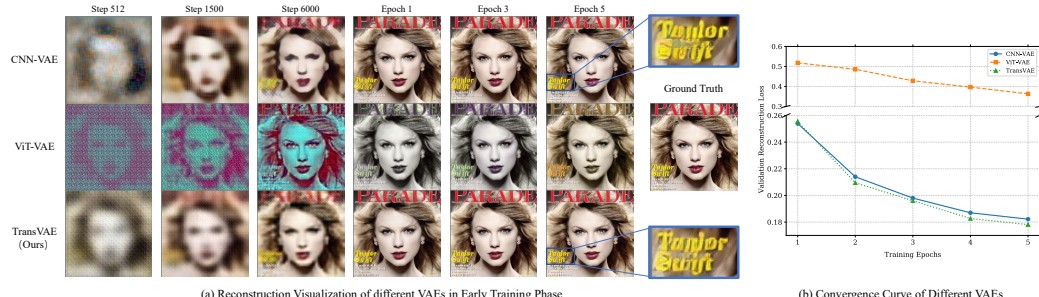

(a) Reconstruction Visualization of different VAEs in Early Training Phase    (b) Convergence Curve of Different VAEs

Figure 1: Comparison of the patterns and loss curves of CNN-VAE, ViT-VAE, and our proposed TransVAE. Our model obtains the local and global representation and superior learning efficiency.

this shared failure to a fundamental limitation: **both paradigms capture an incomplete visual representation**. CNNs are confined to local patterns, while ViTs, without a strong local grounding, struggle to organize their global context meaningfully. Therefore, scaling either in isolation cannot resolve the inherent architectural deficiency.

To break this impasse, we argue that the optimal path forward is not a choice between CNNs and Transformers, but a principled unification of their complementary strengths. In this work, we propose TransVAE, a new hybrid VAE architectural paradigm. TransVAE is designed with a strategic division of labor: it first employs layers of CNN as the front-end to act as an efficient and robust local feature extractor, distilling high-resolution inputs into a compact, yet feature-rich, spatial representation. This representation is then processed by a deep Transformer backbone, which, freed from the burden of learning low-level details, can dedicate its global attention mechanism to modeling long-range semantic dependencies. This synergistic design forms the core principle that allows TransVAE to overcome the limitations of monolithic architectures and capture a complete visual representation.

Extensive experiments validate the efficacy of our proposed paradigm. By capturing a more complete visual representation, TransVAE demonstrates significantly enhanced learning efficiency, converging to high-fidelity reconstruction capabilities much faster than monolithic baselines. Furthermore, this comprehensive representation is the key that unlocks true architectural scalability. We successfully train a family of TransVAE models ranging from 44M to 2.3B parameters and observe a consistent and predictable improvement in performance. Our empirical analysis reveals that scaling TransVAE yields three crucial benefits: (1) superior visual encoding and decoding capabilities, leading to state-of-the-art reconstruction fidelity; (2) enhanced extrapolation, allowing models trained solely on low-resolution data to perform inference on arbitrary higher resolutions with remarkable global coherence; and (3) closer alignment to visual foundation model, resulting in a unified representation that integrates high-level semantic understanding with pixel-level detail, bridging the gap between perception and reconstruction. As a result, our TransVAE achieves a competitive reconstruction performance compared to the modern SD3.5-VAE (Esser et al., 2024) and FLUX-VAE (Labs), while demonstrating a more generation-friendly with a discriminative and disentangled latent space.

Our contributions are two-fold and directly address the challenges outlined above:

- We systematically explore and improve each component of ViT-VAE through comprehensive experiments. We propose TransVAE, a hybrid architecture that unifies a CNN front-end with a Transformer backbone. This design overcomes the "architectural impasse" by capturing a complete visual representation that integrates both local details and global context.
- Scaling from 44M to 2.3B parameters, we show that the complete representation learned by TransVAE gives rise to three critical scaling properties: (1) increasing reconstruction fidelity and (2) enhanced extrapolation, enabling models trained on low resolutions to generalize to arbitrary higher resolutions with remarkable global coherence. (3) better alignment to visual foundation models, harmonizing both pixel-level and semantic-level representation.

## 2 RELATED WORK

Since the era of the Latent Diffusion Model (LDM) (Rombach et al., 2022), VAE has become the core component to accelerate training and inference of diffusion models by compressing the image

from a large spatial scale to a compressed latent space. The widespread VAEs, such as traditional SD-VAE (Rombach et al., 2022), SD3.5 VAE (Esser et al., 2024), and FLUX-VAE (Labs), are built upon the CNN structure, leveraging the strong local prior to compress and extract detailed visual representations, and reconstructing the latents with the decoder with a symmetric structure. Though the ViT (Dosovitskiy et al., 2020) structure has been widely adopted in discrete tokenizers (Yu et al., 2024; Chen et al., 2025b;a; Ma et al., 2025), functioning in auto-regressive generation or unified understanding and generation models, the ViT structure for the continuous variational autoencoder has not been fully explored. ViTok (Hansen-Estruch et al., 2025) is the first to deeply study the potential and scaling property of ViT-based VAE, and reaches a conclusion that the scalability of ViT-VAE is not promising, and it fails to obtain a ViT-VAE with competitive reconstruction performance to CNN-VAE. In addition, DeTok (Yang et al., 2025) is another ViT-based VAE that leverages the latent denoising in training. While it also underperforms normal CNN-VAEs and fails to extrapolate to other resolutions during inference, as shown in Figure 3. MAGI-VAE (Teng et al., 2025) is a video VAE built with pure ViT and also fails in artifacts when inference. Thus, it requires a tiling strategy to constrain its inference grid within a small window. In this paper, we focus on the architecture design of VAE with a hybrid CNN-ViT structure, breaking the common CNN architecture and overcoming the limitations of ViT architectures. With its complete visual representations, our TransVAE demonstrates superior performance in learning efficiency and scalability.

## 3 METHOD

In this section, we elaborate on the proposed TransVAE architecture with a chain of experiments, as evidence to show the effectiveness of the proposed components. First, we introduce the preliminaries about the structure and early learning patterns of traditional CNN-VAE and the ViT-VAE. In addition, we conduct comprehensive experiments to explore how each component contributes to the learning efficiency, gradually transferring from ViT-VAE to our TransVAE, which achieves a complete visual representation with local and global modeling capacity. Finally, with the complete representation, we break the scaling limitation of traditional VAEs of different architectures. With stable training strategies, TransVAE demonstrates an increasing performance on reconstruction ability, inference extrapolation, and the harmony of semantic-level and pixel-level representation.

### 3.1 PRELIMINARIES

In this paper, we mainly focus on the visual autoencoder for image tokenization, specifically the Variational Autoencoder (VAE), because it faces less information loss in the compressed continuous latent space, and provides superior performance over discrete tokenizers in modern generative models. VAE comprises two main components, an encoder $\mathcal{E}$ and a decoder $\mathcal{D}$. Given an input image $\mathbf{x} \in \mathbb{R}^{H \times W \times 3}$, the encoder $\mathcal{E}$ maps the high-dimensional $\mathbf{x}$ to a lower-dimensional latent representation. With Gaussian modeling and KL regularization on the latent representation, VAE builds an information-rich compressed latent space $\mathbf{z} = \mathcal{E}(\mathbf{x}) \in \mathbb{R}^{\frac{H}{f} \times \frac{W}{f} \times d}$, where $f, d$ represent the spatial compression ratio and latent dimension, respectively. Then the decoder reconstructs the latent representation to the original signal $\hat{\mathbf{x}} = \mathcal{D}(\mathbf{z})$ by minimizing the reconstruction loss with $\mathbf{x}$.

The architecture design of the encoder and the decoder is dominated by CNNs, as shown in Figure 2 (a). The CNN-VAE leverages Residual Block (ResBlock) (He et al., 2016) as the basic block of each stage, and uses downsampling and upsampling modules to process the multi-stage features. Some CNN-VAEs (Rombach et al., 2022) adopt the MidBlock, which comprises sandwich-like Conv-Attn-Conv style layers, between the final convolution stage and the latent representation. The Self-Attention block in the MidBlock supports global modeling for CNN-VAE. However, it is found to be less effective in training, as discussed in the Section E.

We first introduce the experimental settings to train VAEs with $f16d32$ for all different structures. We adopt the ImageNet-1k datasets at $256 \times 256$ resolution as the training dataset, with a batch size of 256, base learning rate of $1.0e^{-4}$, and an epoch of 5, resulting in 200k steps in total. We only focus on the reconstruction pattern in Sections 3.1, 3.2, and 3.3, which means our training target is L1 loss for reconstruction, LPIPS loss (Zhang et al., 2018), and KL loss, with a loss weight of 1.0, 1.0, and $1.0e^{-8}$, respectively. For Section 3.4, we adopt the same training strategies, besides training only 3 epochs with a learning rate warmup of 10000 steps, and follow a GAN loss (Esser et al., 2021)

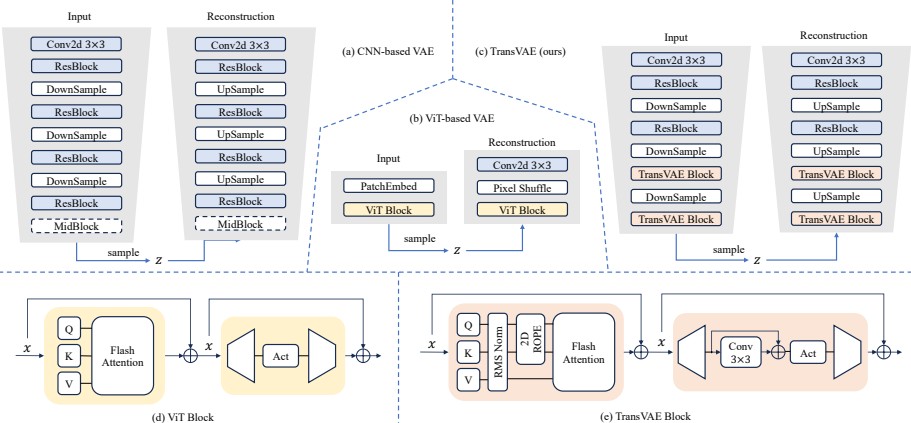

Figure 2: Comparison of CNN-VAE, ViT-VAE, and our proposed TransVAE. Our TransVAE uses stages of a CNN stem for patch embedding and local fine-grained pixel representation, followed by stages of TransVAE blocks to capture global and semantic representations. The TransVAE block equipped with normalization and ROPE can stabilize the large-scale training, and MLP with residual convolution obtains local bias in the deep network. The convolutional projection right before and after the latent representation is ignored for simplicity.

with a weight of 0.05 and VF loss (Yao et al., 2025) with a weight of 0.1 to evaluate the effectiveness of the full training strategy on the scaling models.

We visualize how the CNN-VAE and ViT-VAE behave in the early learning stage in the image reconstruction task, as shown in Figure 1. We can draw two conclusions: (1) CNN-VAE excels in fine-grained local modeling, such as pixel-level color, while ViT-VAE captures coarse-grained global representation, such as the shape and structure. (2) CNN-VAE gradually learns global representation during training but fails in structural details, while ViT-VAE struggles to capture pixel-level color and converges more slowly. We argue that this phenomenon is attributed to the incomplete visual representation of the two structures individually. With a hybrid combination of the two structures, our proposed TransVAE obtains both local and global representation in early training, while also converging faster than CNN-VAE and eliminating the structural artifact. The contribution of each component of TransVAE is elaborated in the following part with insightful experimental evidence.

## 3.2 MACRO DESIGN

We divide the improvements that facilitate the learning convergence of ViT-VAE into two levels: macro designs and micro designs. The macro design includes Position Embedding strategy, Patch Embedding module, and Architectural Staging. The micro design includes the specific designs in the Transformer Block for Self-Attention and Multi-layer Perceptron, and the Upsample or Downsample Module. In this part, we first discuss the macro designs that significantly bridge the convergence gap between the original ViT-VAE and CNN-VAE.

### 3.2.1 RESOLUTION EXTRAPOLATION WITH ROPE

Modern VAEs are capable of generalizing to any resolutions unseen during training. This property, known as extrapolation, is heavily influenced by the model's mechanism for handling spatial information. We analyze the performance of different VAEs when trained on a fixed resolution of $256 \times 256$ and tested directly on a higher resolution of $512 \times 512$. As shown in Figure 3(a), this scenario reveals a critical failure mode in standard ViT-VAEs (Yang et al., 2025; Teng et al., 2025; Hansen-Estruch et al., 2025). These models typically rely on learnable, absolute positional embeddings (APE), which are resolution-specific. When presented with a higher resolution input, the APEs must be interpolated, leading to a disruption of the learned spatial priors. This manifests as severe grid-like artifacts and a catastrophic degradation in reconstruction quality, making the model fail in out-of-distribution resolutions. In contrast, the pure convolutional architecture of CNN-VAE, with its inherent translation equivariance, naturally handles resolution changes without such structural failures.

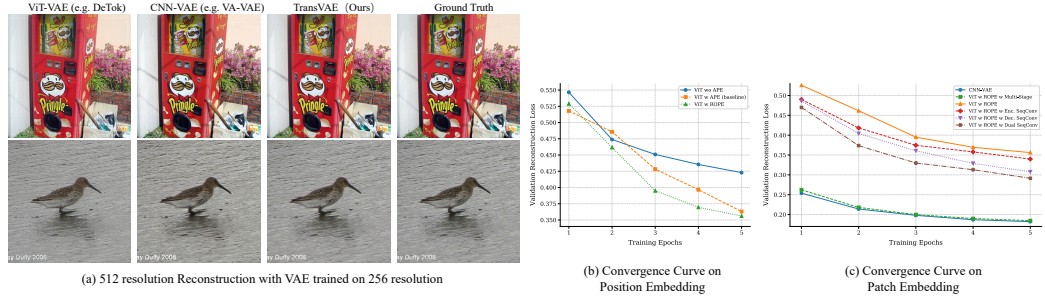

(a) 512 resolution Reconstruction with VAE trained on 256 resolution  (b) Convergence Curve on Position Embedding  (c) Convergence Curve on Patch Embedding

Figure 3: (a) Results of resolution extrapolation during inference for three types of models. (b) and (c) presents the impact of proposed modifications on position embedding and patch embedding.

To mitigate the artifacts, we introduce Rotary Position Embeddings (RoPE) (Su et al., 2024) within the Transformer backbone, to be specific, in the Self-Attention module for Query and Key. Unlike absolute embeddings, RoPE encodes positional information in a relative manner by rotating the feature vectors. This relative formulation is inherently more flexible and robust to changes in sequence length, allowing the model to gracefully extrapolate its understanding of spatial relationships to resolutions far beyond its training domain. Our final TransVAE, equipped with RoPE, not only avoids these failures but also demonstrates superior reconstruction fidelity.

Furthermore, Figure 3(b) shows the convergence curves for models trained with different position embeddings. Removing all position embedding from ViT will degrade the performance, while our application of RoPE not only enhances the training efficiency but also improves generalization. This analysis firmly establishes RoPE as a crucial component for building a truly generalizable and scalable hybrid VAE, and thus it is a core element of our final TransVAE design.

### 3.2.2 FROM FINE-GRAINED PATCH EMBEDDING TO MULTI-STAGE STRUCTURE

While RoPE effectively addresses the challenge of positional generalization, the fundamental weakness of ViTs in modeling fine-grained local details persists, as it is orthogonal to the position embedding strategy. To tackle this, we first explore the architectural choices inherited by typical ViT-VAEs from the original ViT for visual understanding. Standard ViT-VAEs (Hansen-Estruch et al., 2025; Teng et al., 2025; Yang et al., 2025) often employ a non-overlapping convolutional layer for patch embedding and a pixel-shuffle operation for the final output to recover the input spatial resolution, as shown in Figure 2 (b). Such designs, while computationally efficient, tend to treat local patches as independent units or merely shuffle pixels into channel dimensions, largely ignoring the rich spatial correlations within a local neighborhood.

Therefore, our first step is to enhance these local interactions. Inspired by DHVT (Lu et al., 2022) that bridges the gap between CNN and ViT in visual understanding, we replace the standard non-overlapping patch embedding and pixel-shuffle layers with sequential overlapping convolutional layers (SeqConv) for downsampling and upsampling. This design ensures that each downsampling/upsampling step involves a receptive field that overlaps with its neighbors, forcing the model to learn and preserve local spatial structures. The efficacy of this approach is demonstrated in Figure 3 (c), where each application of our sequential convolutional layers yields an acceleration in convergence, confirming their direct contribution to more efficient local detail modeling. Applying Seqconv both for the Encoder and the Decoder greatly enhances the learning efficiency.

Furthermore, inspired by the success of this multi-stage architecture principle in visual representation modeling, we extend this philosophy from component-level to a global architectural redesign. We transform the monolithic, single-stage ViT into a multi-stage hybrid model. Specifically, the initial two stages of our TransVAE architecture are composed of standard CNN blocks, i.e., the ResNet's ResBlocks, which act as a powerful and efficient front-end for hierarchical local feature extraction. The deeper stages then apply traditional ViT blocks, resulting in the "ViT w ROPE w Multi-Stage" variant in Figure 3 (c), which demonstrates a close learning efficiency as CNN-VAE. This hierarchical design allows the model to first build a robust foundation of local features using the inductive biases of CNNs, before leveraging the Transformer's global attention to model high-level semantic

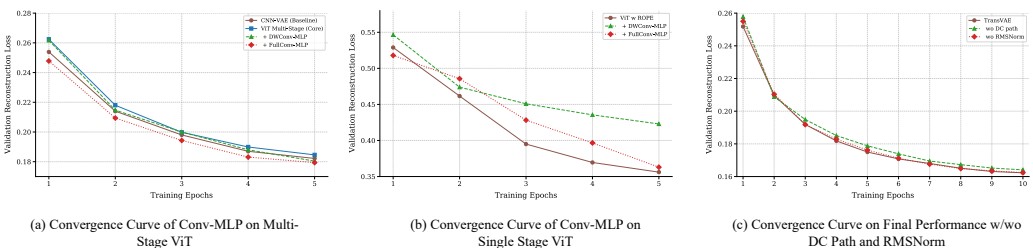

(a) Convergence Curve of Conv-MLP on Multi-Stage ViT

(b) Convergence Curve of Conv-MLP on Single Stage ViT

(c) Convergence Curve on Final Performance w/wo DC Path and RMSNorm

Figure 4: Convergence Curve of Conv-MLP on (a) multi-stage structure and (b) single-stage structure. The DC path accelerates convergence, while RMSNorm has a trivial impact as shown in (c).

relationships. In the next part, we then improve the ViT blocks to TransVAE blocks from a micro-level local prior enhancement perspective, thus creating a truly complete visual representation.

### 3.3 MICRO DESIGN

#### 3.3.1 LOCAL PRIOR ENHANCEMENT WITH CONVOLUTIONAL FFN

While our multi-stage design provides a strong local foundation, a subtle performance gap exists in local detail modeling compared to pure CNNs. We attribute this to the intrinsic structure of the Transformer block itself: its feed-forward network (FFN) operates purely point-wise on each patch token, completely severing local spatial connections at this stage of computation.

To remedy this, we introduce the convolution operation into the standard FFN pipeline, as shown in Figure 2 (e). Specifically, following the "inverted bottleneck" structure of standard FNN, we apply a convolution path to perform spatial mixing for the high-dimensional feature after the input projection. This core step explicitly re-establishes a local inductive bias. A residual connection is added to the convolutional path to maintain the original pointwise projected feature. We investigate two variants of the spatial mixing convolution: a lightweight depth-wise convolution (DWConv) (Sandler et al., 2018), with a $4\times$ channel expansion, inspired by DHVT (Lu et al., 2022), and a more expressive full convolution (FullConv), with a $1\times$ expansion to maintain a comparable parameter budget. The impact of this micro-architectural enhancement is profound. As illustrated in Figure 4 (a) and (b), both variants accelerate training. The DWConv brings the convergence speed to a level that closely approaches the pure CNN baseline, while the FullConv successfully surpasses it. This powerfully demonstrates the benefit of our local prior enhancement design, and we adopt the FullConv version in our final TransVAE architecture.

#### 3.3.2 MINOR MODIFICATIONS

To complete the TransVAE design, we incorporate two final modifications aimed at enhancing training stability at scale and preserving high-resolution fidelity for high spatial compression. First, to ensure stable training as we scale our models to billions of parameters, we introduce a normalization layer immediately before the Q, K, and V linear projections within each self-attention block. For this purpose, we opt for Root Mean Square Normalization (RMSNorm) (Zhang & Sennrich, 2019). RMSNorm simplifies the standard Layer Normalization by re-scaling the activations based only on the root mean square statistic, which has been shown to be both computationally efficient and highly effective in stabilizing the training of large-scale Transformers. Second, to preserve fidelity under high compression ratios, we follow DC-AE (Chen et al., 2025c) to modify all spatial downsampling and upsampling operations with a parallel, information-preserving shortcut. This shortcut leverages convolutional pixel unshuffling/shuffling operations to create a branch alongside the common module. Combining the outputs of both paths ensures that fine-grained details, which might be otherwise lost during aggressive spatial resizing, are faithfully propagated across different stages of the network. As evidenced by a longer training curve Figure 4 (c), introducing such a path can also facilitate convergence. These final modifications, combined with the macro- and micro-architectural designs discussed previously, result in our final TransVAE model with a complete visual representation by harmonizing the local and global modeling capacity. In conclusion, we only adopt the **simplest** modification to build our TransVAE, such as preserving the vanilla self-attention mechanism (though implemented by modern Flash Attention (Dao et al., 2022)), and a slightly modified FFN. We leave a broad design space for future work to develop more efficient architectures.

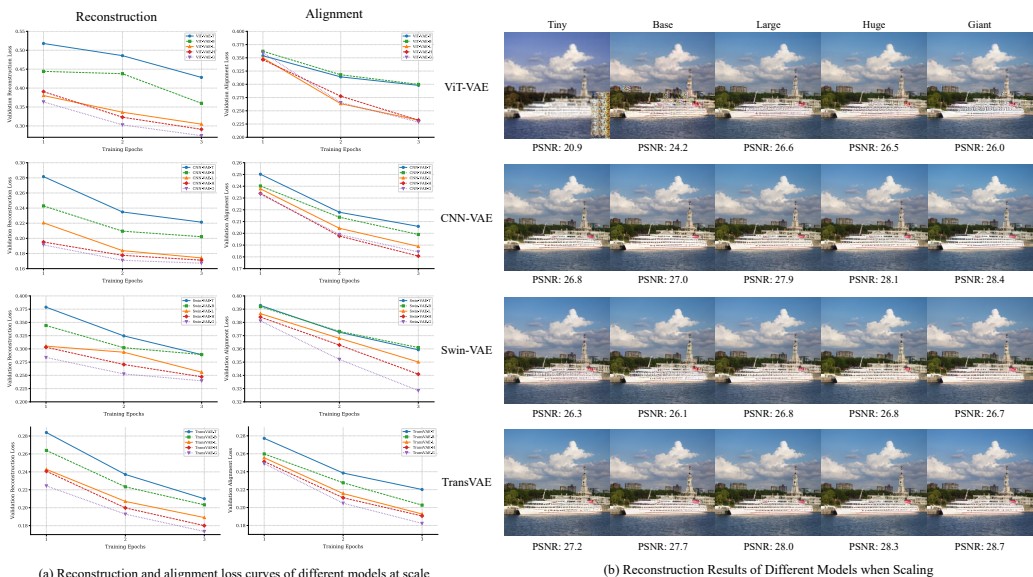

(a) Reconstruction and alignment loss curves of different models at scale

(b) Reconstruction Results of Different Models when Scaling

Figure 5: Comparison of CNN-VAE, ViT-VAE, Swin-VAE, and our proposed TransVAE at scale. "T", "B", "L", "H", and "G" represent Tiny, Base, Large, Huge, and Giant, respectively. (a) We show the validation loss on reconstruction and alignment for models. (b) Visualization results of models trained for 3 epochs, demonstrating the learning efficiency of TransVAE over baselines.

### 3.4 SCALING PROPERTIES OF TRANSVAE

A good scaling property represents that the validation loss can be obviously improved by increasing the model size. In order to examine the effectiveness of TransVAE in the modern VAE training pipeline, we further applied the GAN loss to reduce the artifacts for more realistic reconstructions, and the VF loss in VA-VAE (Yao et al., 2025) to align the latent space with DINOv2 for a more discriminative space and friendly for downstream DiT training. The basic experimental setting is stated in Section 3.1, and the model configuration of the experimental scaling models is demonstrated in the Section C. We further examine the effectiveness of the well-established Swin Transformer (Liu et al., 2021) in the VAE architecture, because it can also combine the local and global representation.

We plot the validation loss curve of reconstruction and alignment in Figure 5 (a) and the visualization results of each model with the corresponding PSNR value in Figure 5 (b). As evidenced by the loss curve, our proposed TransVAE produces clear scaling benefits on both reconstruction and representation alignment tasks. The ViT-VAE requires a substantial amount of parameters to represent the details, and would suffer from high-frequency artifacts as the model size is insufficient to handle the detailed pixel-level representation. Though both validation losses of Swin-VAE converge faster with model scaling, its subtle visual details still underperform and result in low PSNR during inference. The CNN-VAE and our TransVAE are the two models that successfully achieve scalability. Though CNN-VAE has marginally better reconstruction loss than ours, its PSNR still behaves worse, and the reconstruction gain brought by scaling the size of CNN-VAE is limited, as shown by the loss curve and the visualization. Note that it is meaningless to compare the value of alignment loss across different architectures of models, because the VF loss can be simply controlled or loosened by setting a different margin. In conclusion, our TransVAE can truly scale up and bring in profound performances both in the basic reconstruction task of VAE and the representation alignment task to be more downstream-friendly.

## 4 EXPERIMENTS

We have conducted comprehensive experiments to validate the effectiveness of modules in TransVAE in Section 3, which can serve as ablation studies. In this section, we analyse the performance of TransVAE compared to state-of-the-art models across different compression ratios and benchmarks.

Table 1: Comparison on ImageNet Val and MS-COCO Val with different tokenizers. "†" represents that the model is trained on the in-house dataset. All the results are re-implemented under the same setting, except that ViTok's results are directly cited from its paper, as the checkpoint is not released.

| Tokenizer | # Params | Variant | ImageNet-1k Val | | | | MSCOCO2017 Val | | | |
|---|---|---|---|---|---|---|---|---|---|---|
| | | | rFID ↓ | PSNR ↑ | LPIPS ↓ | SSIM ↑ | rFID ↓ | PSNR ↑ | LPIPS ↓ | SSIM ↑ |
| $256 \times 256$ | | | | | | | | | | |
| SD3.5-VAE | 84M | $f8d16$ | 0.19 | 31.29 | 0.060 | 0.88 | 1.67 | 31.17 | 0.056 | 0.89 |
| FLUX-VAE | 84M | $f8d16$ | 0.18 | 32.80 | 0.044 | 0.91 | 1.35 | 32.74 | 0.042 | 0.92 |
| TransVAE-L † (Ours) | 719M | $f8d16$ | 0.68 | 32.89 | 0.042 | 0.92 | 2.68 | 32.85 | 0.039 | 0.92 |
| DeTok-BB-FT | 171M | $f16d16$ | 0.54 | 25.24 | 0.137 | 0.71 | 3.94 | 24.94 | 0.135 | 0.717 |
| ViTok S-B/16 | 129M | $f16d16$ | 0.50 | 24.36 | - | 0.75 | 3.94 | 24.45 | - | 0.76 |
| ViTok S-L/16 | 427M | $f16d16$ | 0.56 | 24.74 | - | 0.76 | 3.97 | 24.82 | - | 0.77 |
| VA-VAE | 70M | $f16d32$ | 0.28 | 27.64 | 0.098 | 0.78 | 2.72 | 27.43 | 0.094 | 0.79 |
| TransVAE-T (Ours) | 44M | $f16d32$ | 0.40 | 28.38 | 0.091 | 0.81 | 3.06 | 28.11 | 0.089 | 0.82 |
| TransVAE-L (Ours) | 545M | $f16d32$ | 0.66 | 28.92 | 0.086 | 0.82 | 3.51 | 28.69 | 0.083 | 0.83 |
| $512 \times 512$ | | | | | | | | | | |
| SD3.5-VAE | 84M | $f8d16$ | 0.11 | 33.54 | 0.060 | 0.91 | 0.93 | 32.56 | 0.063 | 0.90 |
| FLUX-VAE | 84M | $f8d16$ | 0.05 | 35.40 | 0.042 | 0.94 | 0.58 | 34.30 | 0.045 | 0.93 |
| TransVAE-L † (Ours) | 719M | $f8d16$ | 0.09 | 34.54 | 0.053 | 0.93 | 0.85 | 33.85 | 0.053 | 0.93 |
| DeTok-BB-FT | 171M | $f16d16$ | 3.43 | 25.22 | 0.262 | 0.70 | 10.50 | 24.69 | 0.254 | 0.69 |
| ViTok S-B/16 | 129M | $f16d16$ | 0.18 | 24.36 | - | 0.75 | 3.94 | 24.45 | - | 0.76 |
| VA-VAE | 70M | $f16d32$ | 0.24 | 29.38 | 0.119 | 0.80 | 2.40 | 28.49 | 0.120 | 0.79 |
| TransVAE-T (Ours) | 44M | $f16d32$ | 0.27 | 29.66 | 0.115 | 0.82 | 3.04 | 27.63 | 0.146 | 0.79 |
| TransVAE-L (Ours) | 545M | $f16d32$ | 0.32 | 30.12 | 0.110 | 0.83 | 2.27 | 29.07 | 0.113 | 0.82 |
| $1024 \times 1024$ | | | | | | | | | | |
| SD3.5-VAE | 84M | $f8d16$ | 0.03 | 37.61 | 0.034 | 0.97 | 0.31 | 36.80 | 0.038 | 0.97 |
| FLUX-VAE | 84M | $f8d16$ | 0.01 | 39.84 | 0.023 | 0.98 | 0.16 | 39.13 | 0.025 | 0.98 |
| TransVAE-L † (Ours) | 719M | $f8d16$ | 0.02 | 39.65 | 0.30 | 0.98 | 0.28 | 38.90 | 0.045 | 0.97 |
| DeTok-BB-FT | 171M | $f16d16$ | 8.86 | 25.06 | 0.371 | 0.73 | 18.53 | 24.87 | 0.357 | 0.72 |
| VA-VAE | 70M | $f16d32$ | 0.37 | 33.61 | 0.116 | 0.89 | 2.10 | 32.66 | 0.121 | 0.88 |
| TransVAE-T (Ours) | 44M | $f16d32$ | 0.27 | 32.20 | 0.124 | 0.88 | 1.87 | 31.35 | 0.131 | 0.87 |
| TransVAE-L (Ours) | 545M | $f16d32$ | 0.11 | 34.06 | 0.091 | 0.91 | 1.15 | 33.07 | 0.100 | 0.87 |

## 4.1 EXPERIMENTS SETUP

**For visual tokenizer training**, we employ the training strategy following (Rombach et al., 2022; Li et al., 2024; Yao et al., 2025). The main structure follows the VQGAN (Esser et al., 2021) with KL Loss for continuous latent space regularization, while replacing the Encoder and Decoder architecture with our TransVAE. We train two different variants of compression ratio: (1) $f8d16$ variant on the in-house large-scale dataset with roughly 25M images, to compare with SOTA open-source FLUX-VAE and SD3.5-VAE. (2) $f16d32$ variant trained solely on ImageNet-1k dataset, to make a fair comparison with baseline VA-VAE $f16d32$ variant. **For the downstream image generation training**, we also follow the scheme of VA-VAE. Due to the constrained computation resource, we only train at $256 \times 256$ resolution, for LightningDiT-B/2 with $f8d16$ VAEs on ImageNet for 160 epochs, LightningDiT-B/1 for 160 epochs with $f16d32$ VAEs, and LightningDiT-XL/1 for 80 epochs with $f16d32$ VAEs. Detailed experiment settings are elaborated in Appendix Section D.

## 4.2 IMAGE RECONSTRUCTION PERFORMANCE

We adopt the ImageNet-1k validation set and COCO2017 (Lin et al., 2014) validation set as the main benchmarks. We evaluate the models on $256 \times 256$, $512 \times 512$, and $1024 \times 1024$ resolutions. This ensures a robust evaluation of different VAEs across different data distributions and arbitrary resolutions. Note that, except for SD3.5-VAE and FLUX-VAE, all other models are trained on $256 \times 256$ solely. As shown in Table 1, our TransVAE achieves competitive reconstruction performance against different state-of-the-art methods across multiple datasets and resolutions. Both sizes of our TransVAE perform better than VA-VAE on $256 \times 256$ resolution that have been trained on, while our larger size model demonstrates superior performance in higher resolution over all the metrics, providing a scaling property on extrapolation. The baseline DeTok fails to extrapolate, as it adopts absolute position embedding. The other baseline ViTok enables extrapolation with its ROPE design, while its reconstruction performance remains inferior due to the lack of local priors.

## 4.3 DOWNSTREAM TASKS PERFORMANCE

We evaluate the latent space of pretrained VAEs by image generation with LightningDiT. We first train a LightningDiT-B/2 with TransVAE-L-$f8d16$, FLUX-VAE, and SD3.5-VAE. As shown in Figure 6 (a), trained with alignment loss, our TransVAE achieves 37.43 FID at 100k steps, exhibits a $2\times$ faster convergence than FLUX-VAE at 200k steps. We further train LightningDiT-B/1 and

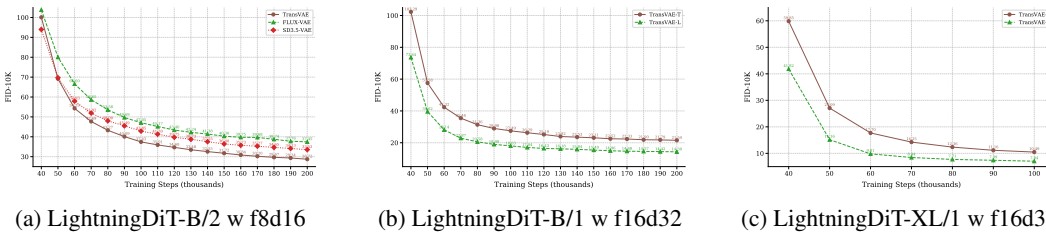

(a) LightningDiT-B/2 w f8d16  (b) LightningDiT-B/1 w f16d32  (c) LightningDiT-XL/1 w f16d32

Figure 6: FID-10k curve (without CFG) of LightningDiT with different VAEs.

LightningDiT-XL/1 with TransVAE-T-$f16d32$ and TransVAE-L-$f16d32$, as shown in Figure 6 (b) and (c). The convergence of the larger TransVAE is faster, with a $2\times$ acceleration, which supports the scaling properties of TransVAE in the downstream tasks. Thus, scaling TransVAE demonstrates a better trade-off between high fidelity reconstruction and downstream-friendly latent space.

## 4.4 INFLUENCE OF VF LOSS

We evaluate the influence of VF loss on VAEs in Table 2. We perform linear probing (Radford et al., 2021) on the latent space for different models on ImageNet-1k dataset, as shown in Table 2 (a). Training without VF loss results in a semantic-entangled latent space, making it hard to discriminative between features. Our TransVAE-L achieves 58.77% accuracy, showing its potential to unify the pixel-level and semantic-level representation. It also outperforms TransVAE-T with 46.61% accuracy, demonstrating the effectiveness of model scaling. For general domain VAEs, our model also shows a more disentangled latent space compared to SD3.5-VAE and FLUX-VAE. This property is also proved by Table 2 (b), which we follow the metric provided by (Yao et al., 2025), showing that our latent space is more discriminative and disentangled. In addition, as evidenced in Table 2 (c), when removing the VF loss in training, the performance upper bound of TransVAE is superior and demonstrates good extrapolation capability. VF loss significantly degrades the reconstruction for both VA-VAE and TransVAE, and the gap is exacerbated on larger resolutions. We hypothesize that the alignment requires training on images of different sizes, which motivates us to adopt multi-resolution reconstruction training and semantic alignment in the future.

Table 2: Influence on VF Loss.

(a) Linear probe accuracy.

| In Domain $f16d32$ VAE | |
| --- | --- |
| TransVAE-T wo VF | 10.91 |
| VAVAE | 45.03 |
| TransVAE-T | 46.61 |
| TransVAE-L | **58.77** |

| General Domain $f8d16$ VAE | |
| --- | --- |
| SD3.5-VAE | 10.48 |
| FLUX-VAE | 7.82 |
| TransVAE-L | **14.87** |

(b) Latent Space Measurement.

| Model | Density CV ↓ | Norm. Entropy ↑ | Gini ↓ | FID-10K ↓ |
| --- | --- | --- | --- | --- |
| SD3.5-VAE | 0.22 | 0.9970 | 0.122 | 33.52 |
| FLUX-VAE | 0.22 | 0.9970 | 0.126 | 37.45 |
| TransVAE-L | **0.20** | **0.9976** | **0.112** | **28.72** |

(c) $f16d32$ Reconstruction metrics.

| Model | Res. | rFID ↓ | PSNR ↑ | LPIPS ↓ | SSIM ↑ |
| --- | --- | --- | --- | --- | --- |
| VAVAE | 256 | 0.28 | 27.96 | 0.096 | 0.79 |
| wo VF | 256 | 0.26 | 28.59 | 0.089 | 0.80 |
| TransVAE-T | 256 | 0.40 | 28.38 | 0.091 | 0.81 |
| wo VF | 256 | 0.34 | 28.94 | 0.091 | 0.82 |
| TransVAE-L | 256 | 0.66 | 28.92 | 0.086 | 0.82 |
| wo VF | 256 | 0.58 | 29.36 | 0.087 | 0.82 |
| TransVAE-L | 512 | 0.32 | 30.12 | 0.110 | 0.83 |
| wo VF | 512 | 0.17 | 31.05 | 0.090 | 0.85 |
| TransVAE-L | 1024 | 0.11 | 34.06 | 0.091 | 0.91 |
| wo VF | 1024 | 0.07 | 36.38 | 0.063 | 0.93 |

## 5 CONCLUSION

In this paper, we break the traditional CNN-VAE architecture with a hybrid CNN-ViT VAE model, TransVAE. With a series of comprehensive explorations and improvements on each component, we overcome the limitation of ViT-VAE and progressively refine its structure to the final TransVAE with a multi-stage locally enhanced design. TransVAE demonstrates better learning efficiency and scalability in reconstruction, extrapolation, and semantic alignment capacity. Experiments support the effectiveness of TransVAE across different benchmarks and resolutions, proving its reconstruction performance, downstream-friendly properties, and scaling benefits. The design space of TransVAE is simple and effective, leaving much space for future research.

ETHIC STATEMENT

This research focuses on developing TransVAE, a high-fidelity neural image codec designed for efficient representation learning and compression, rather than novel image synthesis. The work is purely technical in nature and does not involve the use of sensitive personal data, human subjects, or any form of personally identifiable information.

All models presented in this paper were trained and evaluated exclusively on datasets that are either publicly available (e.g., ImageNet, COCO) or were collected in compliance with all applicable ethical guidelines. We have ensured that any internally curated data is fully anonymized and does not contain any content that would violate ethical principles.

Given that the primary function of our model is high-fidelity reconstruction (acting as a tokenizer), it does not directly generate novel images from random noise and therefore does not present the same risks of misuse (e.g., for creating "deepfakes") as purely generative models like GANs or Diffusion Models. We acknowledge that, as an enabling technology, the high-quality representations produced by TransVAE could potentially be used as input for separate, downstream generative models. However, the ethical considerations of such downstream applications, while important, are distinct from the scope of this work, which is focused on the foundational task of efficient and scalable image tokenization. We hope our contributions will foster positive advancements by providing a more efficient backbone for large-scale vision models, enabling research in areas like visual search, pattern recognition, and serving as a foundational component for future AI systems.

REPRODUCIBLILITY STATEMENT

To ensure the reproducibility of our work, we will release our complete source code and all pre-trained model weights upon publication. Our implementation, based on PyTorch and building upon publicly available codebases, is designed to be clear and self-contained. Our experiments rely on standard public datasets (e.g., ImageNet, COCO) for evaluation, and while some training involved an internal dataset that cannot be released, all key results and ablation studies presented in this paper are on public data, allowing for independent verification of our claims. Comprehensive details regarding the experimental setup are provided in the Appendix, including specific hyperparameter settings, optimizer configurations, data processing pipelines, and training procedures for each model. Furthermore, all models were trained on commercially available H20 GPUs, which can lower the barrier for reproduction without requiring specialized, high-end computational resources.

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

APPENDIX

## A  LLM USAGE STATEMENT

LLM Usage Statement In accordance with ICLR's policy, we disclose that Large Language Models (LLMs) were used as an assistive tool in the preparation of this manuscript. The usage was strictly limited to improving the quality of the writing and assisting with document formatting. Specifically, the LLM was utilized for the following tasks:

- Language and Wording: Improving grammar, rephrasing sentences for clarity, correcting spelling, and refining the overall academic tone of the text.
- Structural Editing: Providing suggestions for structuring paragraphs and enhancing the narrative flow of arguments to improve readability.
- Technical Document Preparation: Assisting with the generation of complex LaTeX code for tables and providing guidance on document formatting.

We emphasize that all core scientific ideas, algorithms, experimental designs, results, and conclusions presented in this paper are the original work of the human authors. The LLM was not used to generate any scientific content or claims. The final text was thoroughly reviewed, edited, and validated by the authors, and we take full responsibility for the entire content of this submission.

## B  MODEL CONFIGURATIONS FOR TRANSVAE VARIANTS

We report the TransVAE model configurations and the inference GPU memory demands, which are measured on H20 GPU with BFloat16 precision. The convolution out projection and in projection neighboring to the latent space is applied with $3 \times 3$ convolution.

Table 3: Model configurations for **TransVAE** variants. "Base Dim" lists the channel dimensions for each stage. We list the configuration of each Encoder stage, and the Decoder adopts the symmetric setting. The MLP ratio is always set to 1. The head dimension for self-attention is always set to 64. The GPU memory is measured during inference on $256 \times 256$ resolution with a batch size of 16 under BFloat16 precision.

| Variant | Ratio | Depth | Base Dim | #Params (Encoder+Decoder) | #Params (Total) | Infer. GPU Mem |
|---------|-------|-------|----------|---------------------------|-----------------|----------------|
| Large | f8d16 | [3,3,6,8] | [192, 384, 768, 1536] | 353M + 366M | 719M | 6.3G |
| Tiny | f16d32 | [3,3,3,3,3] | [128, 128, 256, 256, 512] | 21M + 23M | 44M | 7.4G |
| Base | f16d32 | [3,3,3,3,3] | [128, 128, 256, 512, 1024] | 67M + 73M | 140M | 7.6G |
| Large | f16d32 | [3,3,3,4,6] | [192, 192, 384, 768, 1536] | 266M + 279M | 545M | 12.0G |
| Huge | f16d32 | [3,3,4,6,8] | [256, 256, 512, 1024, 2048] | 633M+658M | 1.3B | 17.0G |
| Giant | f16d32 | [3,3,4,8,10] | [320, 320, 640, 1280, 2560] | 1.1B + 1.2B | 2.3B | 21.5G |

# C    MODEL CONFIGURATIONS FOR SCALING EXPERIMENT

In this part, we provide the configurations of the experimental scaling models. Note that in order to compare with modern CNN-VAEs such as VA-VAE, FLUX-VAE, and SD3.5-VAE, the configuration of CNN adopts attention blocks in some stages and the MidBlocks. We report the model configurations and the inference GPU memory demands, which are measured on H20 GPU with BFloat16 precision.

Table 4: Model configurations for **CNN-VAE** f16d32 variants. "Base Dim" refers to the base channel dimension, and "Ch Mult" is the channel multiplier for each stage. We list the configuration of each Encoder stage, and the Decoder adopts the symmetric setting. The GPU memory is measured during inference on $256 \times 256$ resolution with a batch size of 16 under BFloat16 precision.

| Variant | Depth | Base Dim | Ch Mult | #Params (Encoder + Decoder) | #Params (Total) | Infer. GPU Mem |
|---------|-------|----------|---------|------------------------------|-----------------|----------------|
| Tiny  | [3,3,3,3,3] | 128 | [1,1,2,2,4] | 29M + 41M   | 70M  | 7.3G  |
| Base  | [3,3,3,3,3] | 192 | [1,1,2,2,4] | 64M + 93M   | 157M | 7.4G  |
| Large | [3,3,3,3,3] | 384 | [1,1,2,2,4] | 332M + 371M | 703M | 14.2G |
| Huge  | [3,3,3,3,3] | 512 | [1,1,2,2,4] | 450M + 660M | 1.1B | 18.4G |
| Giant | [3,3,3,3,3] | 768 | [1,1,2,2,4] | 1.0B + 1.5B | 2.5B | 31.1G |

Table 5: Model configurations for **ViT-VAE** f16d32 variants. "Embed Dim" is the dimension of the ViT backbone. The MLP ratio is set to 4.0 for all variants. We list the configuration of each Encoder stage, and the Decoder adopts the symmetric setting. The GPU memory is measured during inference on $256 \times 256$ resolution with a batch size of 16 under BFloat16 precision.

| Variant | Depth | Embed Dim | #Params (Encoder + Decoder) | #Params (Total) | Infer. GPU Mem |
|---------|-------|-----------|------------------------------|-----------------|----------------|
| Tiny  | 8  | 512  | 26M + 25M   | 51M  | 7.2G  |
| Base  | 12 | 768  | 86M + 85M   | 171M | 7.9G  |
| Large | 24 | 1024 | 303M + 302M | 605M | 10.4G |
| Huge  | 24 | 1536 | 681M + 680M | 1.4B | 14.4G |
| Giant | 24 | 2048 | 1.2B + 1.3B | 2.5B | 21.3G |

Table 6: Model configurations for **Swin-VAE** f16d32 variants. "Base Dim" lists the channel dimensions for each stage. The patch size is set to 2 for all variants, i.e., the input for the first stage is a patch size of 2. We list the configuration of each Encoder stage, and the Decoder adopts the symmetric setting. The GPU memory is measured during inference on $256 \times 256$ resolution with a batch size of 16 under BFloat16 precision.

| Variant | Depth | Base Dim | #Params (Encoder+Decoder) | #Params (Total) | Infer. GPU Mem |
|---------|-------|----------|----------------------------|-----------------|----------------|
| Tiny  | [3,3,3,3] | [96, 192, 384, 768]     | 30M + 30M   | 60M  | 7.4G  |
| Base  | [3,3,3,3] | [128, 256, 512, 1024]   | 64M + 93M   | 157M | 7.6G  |
| Large | [3,3,4,6] | [256, 512, 1024, 2048]  | 375M + 375M | 750M | 12.0G |
| Huge  | [3,4,6,8] | [320, 640, 1280, 2560]  | 788M + 788M | 1.6B | 17.0G |
| Giant | [3,4,6,8] | [384, 768, 1536, 3072]  | 1.1B + 1.1B | 2.2B | 21.5G |

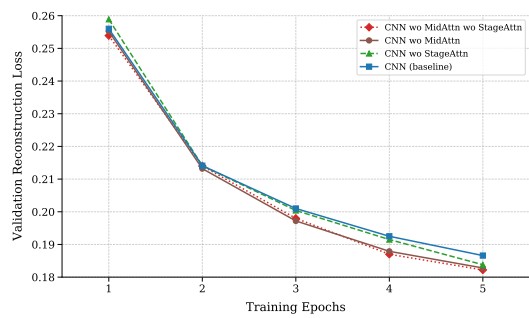

Figure 7: Ablation on the attention in CNN-VAE

## D    DETAILED EXPERIMENT SETTINGS

- For visual tokenizer training, we follow the VA-VAE to adopt the basic training settings: $1.0e^{-4}$ constant base learning rate, 1000 steps of linear warmup, with a global batch size of 256. We use the dynamic loss re-weighting scheme, and the same margin for VF loss to align the latent space to DINOv2 (Oquab et al., 2023). Specifically, we use a weight of 1.0 for L1 reconstruction loss, 1.0 for LPIPS loss, 0.1 for GAN loss, $1.0e^{-8}$ for KL Loss, and 0.1 for VF Loss. There is no weight decay and dropout. Different from the VA-VAE setting that applies three-stage training with L1 Loss, LPIPS Loss, KL Loss, GAN Loss, and VF Loss for 130 epochs in total, we empirically develop a two-stage training strategy. In the first stage, we apply the above losses, other than the GAN Loss, to train both the Encoder and Decoder for 100 epochs. In the second stage, we freeze the Encoder and only train the Decoder with all the losses for 10 epochs. **The training resolution for all the experiments on ImageNet is only $256 \times 256$, without any dynamic resolution or higher resolution training scheme**.

- For downstream image generation training, we train three different variants of LightningDiT on $256 \times 256$ resolution. We extract latent features from the pretrained VAEs to accelerate generation training. We strictly follow the VA-VAE setting for training and inference on the image generation task, while we only change the VAEs from VA-VAE to SD3.5-VAE, FLUX-VAE, and our TransVAE. The LightningDiT-B/2 is trained with SD3.5-VAE, FLUX-VAE, and our TransVAE-L-$f8d16$ for 160 epochs. The LightningDiT-B/1 and LightningDiT-XL/1 are trained with TransVAE-T-$f16d32$ and TransVAE-L-$f16d32$ for 80 and 160 epochs, respectively. The global batch size is 1024, with a constant learning rate of 0.0002 and AdamW (Loshchilov, 2017) beta2 of 0.95.

## E    DISCUSSION OF ATTENTION IN CNN-VAE

As mentioned in the Section 3.1, we empirically find that removing the attention layer in CNN-VAE can improve convergence. We use "StageAttn" and "MidAttn" to denote the attention layer in the CNN stage and the MidBlock, respectively. As shown in Figure 7, removing either of the attention layers can accelerate convergence. After removing all the attention layers, CNN-VAE achieves the optimal convergence speed. Thus, we adopt such a variant as the CNN-VAE baseline for our Method Section.

## F    HOW TO ALIGN WITH VISION FOUNDATION MODEL

In the original VA-VAE, the alignment is seamless on the spatial scale. Because the DINOv2 has a patch embedding of 14, projecting the input $224 \times 224$ into $16 \times 16$ feature map, corresponding to the $256 \times 256$ to $16 \times 16$ of $f16$ VAE variants. However, when we want to align the $f8$ VAE to the DINOv2, the $32 \times 32$ latent space is not compatible with the DINOv2 feature map. There are three solutions: (1) Downsample the VAE latents, (2) Upsample the DINOv2 feature map, (3) Use a large size image, $448 \times 448$. We empirically find that the second solution makes it hard to converge on VF loss, and the first solution is able to converge as normal. It is also compatible with the

intuition that the semantic representation is global, thus downsampling the latents to feature map size is reasonable. The third solution is also reasonable; however, due to its extreme burden in training, we do not experiment with such a variant.

# G    POTENTIAL OF ASYMMETRIC ENCODER AND DECODER

Our TransVAE and traditional CNN-VAEs adopt a symmetric design for the Encoder and Decoder. The parameter of the Decoder is commonly larger than Encoder, while they are the same network structure. Thus, we raise a question: can we devise an asymmetric VAE that uses different structures for the Encoder and the Decoder?

In our initial design of TransVAE, we adopt the Transformer-based Encoder and CNN-based Decoder. Such a design achieves satisfactory reconstruction performance, and it even surpasses the final version of our TransVAE when the model size is at the Tiny level, roughly 60M parameters. This is reasonable because the CNN structure is efficient in modeling pixel-level representation and decoding it with subtle details. However, when we want to scale up, this asymmetric model is quite unstable during training. The training loss is easier to become Nan even when all the stabilizing strategy is introduced: weight initialization, more stable normalization, gradient clip, and very small learning rate, etc.

When we transfer back to the symmetric design, the training becomes stable again, and we can scale the model to even 2.3B parameters without any unstable circumstances. Thus, currently, we reach a conclusion that the asymmetric architecture of the Encoder and Decoder can not scale up. We will leave this question for future, deeper research.

# H    DISCUSSION BETWEEN TRANSVAE AND DC-AE

DC-AE (Chen et al., 2024) is a popular VAE structure that is designed for very high compression ratios, with its "lossless" path design in the upsample/downsample module. We also adopt such a design to facilitate training. DC-AE replaces the attention blocks of traditional CNN-VAE with its EfficientViT (Cai et al., 2023) blocks. Thus, the main structure of DC-AE and our TransVAE is similar.

However, the two structures are different considering the following points. First, the core structure of DC-AE is still convolution blocks. For example, its model configuration is "[0,4,8,2,2,2]" for Encoder and "[0,5,10,2,2,2]" for Decoder, which means each deeper stage with EfficientViT has only 2 blocks. The convolution blocks are still overwhelmed. While in our TransVAE configuration model, we adopt more TransVAE blocks than convolution blocks, as shown in Table 3.

Second, we adopt simple Flash-Attention with QKV Norm and ROPE, and slightly modified Conv-MLP as the TransVAE block, while DC-AE adopts the EfficientViT block, with a new attention mechanism and mobile convolution as the FFN.

