# OpenReview forum: "A Hybrid Paradigm for Vision Autoencoders: Unifying CNNs and Transformers for Learning Efficiency and Scalability"
_ICLR.cc/2026/Conference — ICLR 2026 Conference Withdrawn Submission_

### Official Review · Reviewer_Fmmx · 2025-11-01

**Soundness:** 3
**Presentation:** 2
**Contribution:** 1
**Rating:** 2
**Confidence:** 4

**Summary:**

This paper proposes TransVAE, a hybrid framework that combines Vision Transformer (ViT) and a Convolutional Neural Network (CNN). It first captures local features from images using a CNN and then a Transformer backbone is followed for global context modeling. As a result, TransVAE showed high-fidelity reconstruction capabilities much faster than existing works. The performance of the proposed model is reported with varying scales from 44M to 2.3B parameters, and validated on two representative benchmarks, ImageNet and COCO.

**Strengths:**

1)	The motivation of paper is easy to follow, with clear and straightforward structure. The proposed TransVAE leverages the complementary strengths of CNN and ViT, and detailed comparisons with CNN-VAE and ViT-VAE—supported by quantitative evidence such as convergence curves—clearly illustrate the relative advantages and weaknesses of each approach.
2)	Extensive experiments were performed on multiple datasets with diverse metrics to demonstrate generalizability of the proposed method’s reconstruction quality.

**Weaknesses:**

1)	Lack of technical novelty. The proposed method does not introduce a fundamentally new learning mechanism but rather applies existing techniques (RoPE [Su et al., 2024], SeqConv [Lu et al., 2022]) with only marginal changes to the existing CNN-VAE and ViT-VAE architectures.
2)	Overall, the details of the proposed method (e.g., RoPE, SeqConv) are difficult to understand since there is no single formula with any definition of the model’s inputs and outputs. Providing explicit mathematical formulations and clear variable definitions would greatly help readers understand the underlying mechanism and reproducibility of the method.
3)	The proposed method handles the limitation of the absolute positional embedding (APE) of ViT-VAE by introducing RoPE. However, without a background knowledge on this previous APE method, it is difficult to understand the differences between these approaches, as the paper provides no detailed explanation of APE or direct comparison between APE and RoPE. Including a side-by-side mathematical comparison between these key elements would help contextualize the novelty and clarify the technical advancement of the proposed approach.

**Questions:**

As stated in line 240, the RoPE is a core component of the proposed method; however, the details of how the RoPE rotates feature vectors to encode positional information in a relative manner are not provided, which makes it difficult to understand how this mechanism contributes to the model’s performance. It would be helpful if the authors could include more detailed mathematical explanations, such as the rotation angles and how the feature vectors being rotated are constructed. Similarly, the details of SeqConv, such as the range of overlapping convolutional layers and the direction of overlapping, are not clearly described. Providing explicit architectural definitions or visual illustrations would greatly improve the clarity.

---

### Official Review · Reviewer_eTdp · 2025-11-03

**Soundness:** 2
**Presentation:** 3
**Contribution:** 2
**Rating:** 4
**Confidence:** 3

**Summary:**

This paper introduces TransVAE, a hybrid visual VAE combining a CNN front-end for local detail extraction and a Transformer backbone for global context modeling. The design resolves the inefficiencies of pure CNN- or ViT-based VAEs by capturing a complete visual representation. The core contributions are the systematic design of this hybrid model, incorporating elements like Rotary Position Embeddings, multi-stage convolutional embedding, and convolution-enhanced FFNs. Experiments show faster convergence, better reconstruction, and strong scaling from 44M to 2.3B parameters. Overall, TransVAE sets a new standard for scalable, high-fidelity visual tokenization.

**Strengths:**

The paper provides a thorough investigation of a hybrid design that integrates RoPE, multi‑stage convolutional embeddings, and convolution‑enhanced FFNs. The analysis is detailed, with Figures 2–4 effectively illustrating the architectural rationale and the interactions among CNN‑VAE, ViT‑VAE, and TransVAE variants.

**Weaknesses:**

1. The paper claims to "break this impasse" between CNNs and Transformers, but the concept of hybrid CNN-Transformer models is well-established in computer vision. The paper mentions DHVT as an inspiration but does not sufficiently differentiate its architectural contributions from such prior hybrid designs, which also aim to combine local and global modeling.
2. The claim of being the first to "break the common CNN architecture" in VAEs is overstated, as the paper itself cites pure ViT-based VAEs like ViTok and DeTok, as well as CNN‑ViT hybrids like *DC‑AE*
3. The application of RoPE to enable resolution extrapolation is a direct and effective transfer of technology from other domains (e.g., LLMs). The paper does not explore or compare against alternative relative positional encoding schemes, making this design choice feel more like a straightforward implementation than a deep exploration.
4. The paper claims that introducing RMSNorm is for "stable training as we scale our models to billions of parameters" (Section 3.3.2). However, the experiment shown in Figure 4(c) demonstrates only a negligible effect on convergence. This inconsistency suggests that the inclusion of RMSNorm lacks empirical support for the claimed purpose.
5. A key set of results for the TransVAE-L f8d16 model relies on a private "in-house large-scale dataset" of 25M images. No details regarding the dataset's content, domain, or quality are provided. This lack of transparency severely hinders the reproducibility of these state-of-the-art claims.

**Questions:**

Please refer to "weakness" for details.

---

### Official Review · Reviewer_agCP · 2025-11-03

**Soundness:** 3
**Presentation:** 3
**Contribution:** 3
**Rating:** 4
**Confidence:** 4

**Summary:**

This paper presents a hybrid framework, TransVAE, which combines the complementary strengths of convolutional neural networks (CNNs) and vison transformers (ViTs) for vision autoencoding. The author(s) motivate TransVAE by highlighting the strengths weaknesses of  CNNs and ViTs for vision autoencoding. This paper argues that CNNs are good at extracting hierarchical features from pixel level data due to their locality and translation equivariance but struggle with capturing long-range dependencies. ViTs on the other hand have global receptive fields that are suitable for global context modelling but fail to capture local dependencies. The new TransVAE combines the strengths of these two architectures (CNNs and ViTs).  TransVAE consists of two sub-units. (1) It uses layers of CNNs to extract robust local features and  maps high-resolution into a compact spatial representation. (2) It uses a type transformer backbone for ViTs to model long-range semantic dependencies.  This paper further presents experimental results that demonstrate the efficiency and scalability of TransVAE.

**Strengths:**

This paper is well-organized and the main contributions are well presented. Section 3 does a good job of describing the components of TransVAE and their respective functions. It also provides well-explained experimental result to illustrate the effectiveness of various design strategies that the TransVAE framework uses.

**Weaknesses:**

1). Missing citations on hybrid CNN and ViT frameworks: There are prior works that have explored hybrid combinations of ViTs and CNNs for image-related tasks.  Section 2 on related works does not provide any information about such hybrid approach. Instead, it focuses on VAEs with CNNs and VAEs with ViTs.  I recommend adding a brief discussion of existing CNN+ViT hybrid frameworks to better understand the contribution of this work. I did a quick search on Google Scholar with this query "Hybrid ViT and CNN for images" and it returned several relevant papers. Including a few of these in the related work section would strengthen the paper’s positioning and demonstrate awareness of broader architectural trend.

2). Computational cost and comparison: Table 1 compares the TransVAE with other models. I noticed that for the f8d16 compression variant, the TransVAE slightly outperforms SD3.5-VAE and FLUX-VAE.  However, this slight improvement comes with a significant increase in model size.  The TransVAE-L uses 719M parameters compared to just 84M each for the SD3.5-VAE and the FLUX-VAE. Could the author(s) clarify whether the "L" or "T" variants of TransVAE can be adapted for f8d16 compression to reduce the parameter footprint while maintaining competitive performance??

Also, the paper evaluates convergence speed based on the number of training iterations or epochs. While this is informative, I recommend including compute time per iteration or epoch to provide a more complete picture of training efficiency. This would help readers assess the practical trade-offs between model complexity and training speed.

3).  Clarification on alignment to visual foundation models: Lines  085-089 suggest that one of TransVAE 's benefit is that it aligns closely with visual foundational models. Lines 101-102 also suggest this. It not clear to me which aspect of the experimental results supports this assertion. I recommend clarifying how the reported metrics or experiments demonstrate this alignment.

**Questions:**

Please consider the three points raises under the weaknesses.

---

### Official Review · Reviewer_U6zA · 2025-11-04

**Soundness:** 3
**Presentation:** 3
**Contribution:** 3
**Rating:** 6
**Confidence:** 4

**Summary:**

This paper introduces TransVAE, a novel hybrid architecture for visual Variational Autoencoders (VAEs) that combines the strengths of Convolutional Neural Networks (CNNs) and Vision Transformers (ViTs). Traditional CNN-based VAEs excel at modeling local details but struggle with long-range dependencies, while ViT-based VAEs offer global context modeling but are inefficient at capturing fine-grained local features. TransVAE addresses this architectural impasse by using a shallow CNN front-end for robust local feature extraction, followed by a deep Transformer backbone for global context modeling. The hybrid design leads to superior learning efficiency, faster convergence, and state-of-the-art results in visual representation learning. Notably, TransVAE demonstrates scalability (performance improves as model size increases), enhanced extrapolation (models trained on low resolutions generalize well to higher resolutions), and harmonization of pixel-level and semantic-level representations, making it effective for both reconstruction and generation tasks. Extensive experiments show that TransVAE outperforms pure CNN and ViT VAEs in reconstruction fidelity, scalability, and downstream task performance, providing a blueprint for next-generation visual VAEs. However it would be great to see some more results from text to image generation systems using this framework.

**Strengths:**

1) Principled Hybrid Architecture:
TransVAE unifies CNNs and Transformers, leveraging CNNs for local feature extraction and Transformers for global context modeling.
2)  The hybrid design enables faster convergence and consistent performance improvements as the model scales from 44M to 2.3B parameters—a feat not achieved by pure ViT VAEs. TransVAE demonstrates predictable scaling benefits in both reconstruction and representation alignment tasks.
3) TransVAE models trained on low-resolution images can perform inference on higher resolutions with superior global coherence, by using Rotary Position Embeddings (RoPE) and multi-stage architectural design.
4) Harmonization of Pixel-Level and Semantic-Level Representations: The architecture facilitates a unified representation that integrates high-level semantic understanding with pixel-level detail, bridging the gap between perception and reconstruction. This leads to state-of-the-art reconstruction fidelity and a more discriminative, disentangled latent space, which is beneficial for downstream generative tasks.

**Weaknesses:**

Weakness:
1) Additional overhead during training and inference time? It’s not clear to me how much additional computational costs are required by making these changes in the architecture.
2) Also the current Setup has only been done on ImageNet, where the numbers are kind of saturated; can we repeat experiments on cc-12m? Which is larger and can show more scaling behavior specially with regards to dataset as was done in MUSE[1].
3) The results will also become stronger if they are presented with text to image benchmarks like MUSE[1].
4) Another point of discussion should be whether this approach works well for MAR[2] kind of approaches which use continuous tokens instead of discrete tokens.


References:
[1]MUSE: Muse: Text-To-Image Generation via Masked Generative Transformers
[2] MAR Autoregressive Image Generation without Vector Quantization

**Questions:**

Overall the paper looks good, but there are few more experiments that should be added to the paper for ex. text to image generation / results with continuous tokenizers etc.

---

### Note · Authors · 2025-11-12

I have read and agree with the venue's withdrawal policy on behalf of myself and my co-authors.